# Left ventricular systolic function impairment in children after balloon valvuloplasty for congenital aortic stenosis assessed by 2D speckle tracking echocardiography

Krzysztof Godlewski[1]ʘ, Paweł Dryżek[2]‡, Elżbieta Sadurska[3]‡, Bożena Werner[4]ʘ*

**1** Department of Paediatric Cardiology, University Clinic Centre of the Medical University of Warsaw, Warsaw, Poland, **2** Department of Cardiology, Polish Mother's Memorial Hospital, Research Institute, Łódź, Poland, **3** Department of Paediatric Cardiology, Medical University of Lublin, Lublin, Poland, **4** Department of Paediatric Cardiology and General Paediatrics, Medical University of Warsaw, Warsaw, Poland

ʘ These authors contributed equally to this work.
‡ These authors also contributed equally to this work.
* bozena.werner@wum.edu.pl

**Data Availability Statement:** All relevant data are within the manuscript and its Supporting Information files.

## Abstract

### Aims

The aim of the study was to evaluate left ventricular (LV) remodeling and systolic function using two-dimensional speckle tracking echocardiographic (2D STE) imaging in children at a long-term (more than 36 months, 107.5±57.8 months) after balloon valvuloplasty for aortic stenosis (BAV).

### Methods and results

40 patients (mean age 9,68 years, 75% male) after BAV and 62 control subjects matched to the age and heart rate were prospectively evaluated. The 2D STE assessment of LV longitudinal and circumferential strain and strain rate was performed. Left ventricular eccentric hypertrophy (LVEH) was diagnosed in 75% of patients in the study group. Left ventricular ejection fraction (LVEF) was normal in all patients. In study group, global longitudinal strain (GLS), global longitudinal strain rate (GLSr) were significantly lower compared with the controls: GLS (-19.7±2.22% vs. -22.3±1.5%, $P < 0.001$), GLSr (-0.89±0.15/s vs. -1.04 ±0.12/s, $P < 0.001$). Regional (basal, middle and apical segments) strain and strain rate were also lower compared with control group. Global circumferential strain (GCS), global circumferential strain rate (GCSr) as well as regional (basal, middle and apical segments) strain and strain rate were normal. Multivariable logistic regression analysis included: instantaneous peak systolic Doppler gradient across aortic valve ($PG_{max}$), grade of aortic regurgitation (AR), left ventricular mass index (LVMI), left ventricular relative wall thickness (LVRWT), left ventricular end-diastolic diameter (LVEDd), peak systolic mitral annular velocity of the septal and lateral corner (S'spt, S'lat), LVEF before BAV and time after BAV and showed that the only predictor of reduced GLS was LV eccentric hypertrophy [odds ratio 6.9; (95% CI: 1.37–12.5), $P = 0.045$].

**Funding:** The authors received no specific funding for this work.

**Competing interests:** The authors have declared that no competing interests exist.

## Conclusion

Patients at long-term observation after BAV present the subclinical LV systolic impairment, which is associated with the presence of its remodeling. Longitudinal deformation is the most sensitive marker of LV systolic impairment in this group of patients.

## Introduction

In many centres, percutaneous balloon aortic valvuloplasty (BAV) is the accepted first-line treatment for congenital aortic stenosis (AS) in the paediatric population. A reduction in pressure gradient between the left ventricle (LV) and the aorta (Ao) is an expected and favourable outcome of this procedure. Dilatation of the stenosed valve, however, quite commonly produces varying degrees of aortic regurgitation (AR). In some of these patients, the clinical problem is the gradual progression of AR, which eventually becomes the main reason for surgical intervention [1–3]. While often producing few symptoms or none at all, residual AS and AR may still adversely affect the left ventricle and lead to its dysfunction and decompensation.

The study by Capoulade et al. [4] presenting the relationship between a reduced ejection fraction and higher mortality in adult patients with aortic valve stenosis, indicates that decreased EF is a marker of advanced stage of the disease. It is therefore justified to employ more sensitive methods of assessing myocardial function, as these would enable the detection of subclinical changes that reflect incipient abnormalities and would allow to prevent advanced or even irreversible changes from developing. Novel echocardiographic techniques, including two-dimensional speckle-tracking echocardiography (2D STE) for assessing myocardial function, enable early detection of myocardial disease [5,6].

The aim of our study was to evaluate LV remodelling and systolic function using 2D STE in children during long-term follow-up after BAV for AS.

## Methods

The study was approved by. The study was approved by the Bioethical Committee of the Medical University of Warsaw (KB/45/2014) and followed the rules and principles of the Helsinki Declaration. All parents or legal guardians as well as patients aged 16 years and older gave their informed written consent.

### Study population

A total of 40 post-BAV patients at the mean age of 9.68±4.73 years were included in our study. Only patients who had undergone BAV more than 3 years before were included, with the mean duration of the post-BAV period being 107.5±57.8 months (range: 37 to 216 months). Patients were excluded from this group if they had a history of aortic valve surgery and/or had other forms of congenital heart disease, cardiac arrhythmias, hypertension, metabolic disorders or genetic diseases. The control group consisted of 62 healthy children (volunteers) aged 3 to 18.03 years with the mean age of 9.28 years matched to the age, gender, body weight, body surface area and heart rate of the study group.

The control group did not differ significantly from the interventional group in terms of age, body weight, body surface area and heart rate (Table 1).

**Table 1. Clinical characteristic of the study population, conventional and tissue Doppler echocardiographic measurements of left ventricle.**

| Parameter | Patients (n = 40) | Controls (n = 62) | P value |
|---|---|---|---|
| Age [years] | 9.7±4.73 | 9.3±4.04 | 0.65 |
| Gender ♂/♀, n | 30/10 | 36/26 | 0.62 |
| Body mass, [kg] | 35.8±19.88 | 34.4±17.24 | 0.70 |
| BSA [m $^2$] | 1.1±0.44 | 1.1±0.37 | 0.99 |
| HR [bpm] | 83.1±12.13 | 79.7±11.92 | 0.16 |
| IVSd [cm] | 0.70±0.16 | 0.59±0.11 | <0.001 |
| LVPWd [cm] | 0.69±0.17 | 0.55±0.1 | <0.001 |
| LVEDd [cm] | 4.6±0.73 | 4.1±0.52 | <0.001 |
| LVEDd Z–score | 2.3±2.06 | 0.6±0.71 | <0.001 |
| LVMI [g/m$^{2,7}$] | 50.0±18.43 | 29.5±6.31 | <0.001 |
| LVRWT | 0.30±0.05 | 0.27±0.02 | 0.003 |
| LVEF [%] | 64.1±2.68 | 63.4±1.75 | 0.14 |
| S' spt [cm/s] | 7.0±1.21 | 7.5±1.04 | 0.06 |
| S' lat [cm/s] | 8.1±2.12 | 9.4±1.7 | 0.001 |

Data expressed as mean±SD or number.

IVSd, interventricular septum end-diastolic diameter; LVPWd, left ventricular posterior wall end-diastolic diameter; LV, left ventricle; LVEDd, left ventricular end-diastolic diameter; LVMI, left ventricular mass index; LVRWT, left ventricular relative wall thickness; LVEF, left ventricular ejection fraction; S'spt, S'lat, peak systolic mitral annular velocity of the septal and lateral corner.

## Echocardiographic assessment

The LV and its function were assessed in the post-BAV patients and compared with the control group. All of the echocardiograms were performed following of the recomendations of the European Association of Cardiovascular Imaging and the American Society of Echocardiography using the iE33 Matrix Ultrasound System (Philips, Koninklijke, Netherlands) with 3–8 MHz and 5 MHz sector probes [7,8].

## Conventional and tissue Doppler measurements

LV end-diastolic diameters (LVEDd) and LV wall thickness were measured from the parasternal long-axis 2D view. LV mass (LVM) was calculated according to the Devereux formula [9]. The left ventricular mass index (LVMI) was determined according to the guidelines described by de Simone [10]. LVMI exceeding 95 centile, (value 38.6 g/m $^{2,7}$) and LV relative wall thickness (LVRWT) <0.44 was considered to be LV eccentric hypertrophy.

By using continuous-wave Doppler (CW-Doppler) and color Doppler an assessment of the LV-Ao instantaneous peak systolic Doppler gradient ($PG_{max}$) and the grade AR (on a 4-point grading scale) was made.

Pulsed-wave TDI (PW-TDI) was used to measure peak systolic velocities of the septal (S'spt), and lateral (S'lat) corner of the mitral annulus in apical four-chamber view (4CH). LVEF was calculated by the Simpson method from apical 4- and 2CH views.

## Two-Dimensional Speckle Tracking Analyses (2D STE)

The 2D STE analysis was performed by one physician (K.G.) using software (QLab version 8.1, Philips) on standard 2D greyscale images from apical 4CH, 3CH, and 2CH views for LV longitudinal strain (LS) and longitudinal strain rate (LSr). PSAX view at the basal, middle and apical

levels was used to assess LV circumferential strain (CS) and circumferential strain rate (CSr). Each strain was measured at end-systole at the moment of aortic valve closure. Frame rate ranged from 70 to 80 frames/s. Sedative drugs were not used. The endocardial border was manually traced at end-systole and the width of the region of interest adjusted to include the whole myocardial wall. Then, the software automatically tracked and accepted segments of good tracking quality, automatically rejecting poorly tracked segments. Tracking quality was visually checked, corrected when needed, and only good quality images were accepted by the operator. Strain values of LV were assessed automatically in 18 segments and then averaged manually by the operator as mean value of each strain and strain rate and presented as regional (basal, middle and apical segments), global longitudinal strain (GLS), global circumferential strain (GCS) as well as global longitudinal strain rate (GLSr) and global circumferential strain rate (GCSr). Segment names and their locations specified by the American and European Society of Echocardiography were used [7].

There are no established normal ranges of left ventricular strain using 2D STE for the paediatric population utilizing commercially available QLab software, Philips. The cut-off values of abnormal GLS, GCS, GLSr and GCSr were calculated from the control group as the mean value - 2SD.

The less negative GLS, GCS, GLSr and GCSr values means the lower systolic function.

For intra- and interobserver variability, echocardiographic measurements in 12 randomly selected patients were repeated. The measurements were performed on the same cardiac cycle. Intra- and interobserver variability was assessed using intraclass correlation and absolute difference between the investigators' measurements divided by the average value of both measurements expressed as percentage.

## Statistical analysis

Continuous variables were presented as the mean±SD and compared with Student's t-test. Categorical variables were presented as percentages and compared with two-tailed Fisher's exact test. The Kruskal-Wallis ANOVA test was used for multiple group comparison. The correlations between continuous variables were assessed with the Pearson correlation coefficient (r) and between categorical variables with the Spearman rank correlation coefficient. A $P$ value lower than 0.05 was considered significant. Multivariable logistic regression analysis was performed to identify predictors of abnormal GLS in the study group. Odds ratios (OR) with 95% confidence interval were calculated. The statistical calculations were performed using Statistica software, version 13.1 (Tibco, Palo Alto, CA) and PQStat software, version 1.8.0. (StatSoft).

## Results

### Clinical data

The clinical characteristics of the study group is given in Table 1 and Fig 1. In the study group, BAV had been conducted during the neonatal period in 26 patients, during infancy in 12 patients, and beyond the age of 1 year in 2 patients.

### Conventional echocardiography and TDI

**Assessment of LV hypertrophy and geometry.**   Left ventricular eccentric hypertrophy (LVEH) was identified in 75% of the patients in the study group (Table 1). The subgroup of patients with LVEH significantly differed from the non-LVEH subgroup in terms of mean $PG_{max}$ ($P = 0.02$), severity of AR ($P = 0.02$) and LVEDd Z-score ($P<0.001$). In terms of the other parameters given in Table 2, including LVEF, no significant differences between these

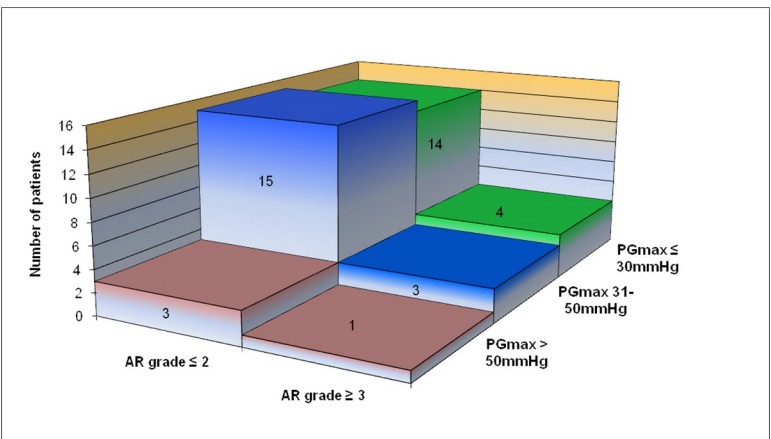

**Fig 1. Characteristic of the study group in regards to the occurrence of PG$_{max}$ and AS.** Abbreviations–see in Table 2.

subgroups were observed. The presence of LVEH significantly correlated with PG$_{max}$ (r = 0.36, $P$ = 0.02), severity of AR (r = 0.65, $P<0.001$) and LVEDd Z-score (r = 0.52, $P<0.001$).

  **Assessment of left ventricular systolic function.**   LVEF was within normal limits in all the study patients and did not differ significantly compared with the control group ($P$ = 0.2). The values of s' spt and s' lat were lower in the study group but the difference was statistically significant only for s' lat ($P$ = 0.001) (Table 1).

## 2D STE Analysis

Speckle tracking was possible in 99.5% of the segments in the apical views (2-, 3- and 4-CH) and in 98.7% of the segments in the short axis LV views (at the levels of the basal, middle and apical parts of the LV).

  The lower limits of normal GLS, GCS, GLSr and GCSr were established in the manner stated in the methodology. The value greater than –19.31%, –19.93%, –0.8/s, and –1.01/s means reduced GLS, GCS, GLSr and GCSr respectively.

  **Analysis of longitudinal strain.**   We found LV systolic function to be less in the group of post-BAV patients compared with the control group. GLS and strain in the basal, middle and

**Table 2. Comparison of selected parameters in subgroups: With and without left ventricular eccentric hypertrophy.**

| Parameter | Group with LVEH (n = 30) | Group without LVEH (n = 10) | *P* value |
|---|---|---|---|
| Age at the time of the echocardiographic examination [months] | 110.9±54 | 126.8±71.5 | 0.46 |
| LVEF [%] (Simpson's method) | 64.4±2.53 | 63.1±3.03 | 0.20 |
| S' spt [cm/s] | 7.1±1.3 | 6.8±0.91 | 0.57 |
| S' lat [cm/s] | 8.2±2.36 | 7.7±1.15 | 0.47 |
| PG$_{max}$ [mmHg] | 35.9±11.9 | 26.1±7.78 | 0.02 |
| AR grade $\geq$ 2/AR grade $<$ 2 | 90%/10% | 20%/80% | 0.02 |
| LVEDd Z–score | 2.9±1.98 | 0.4±0.8 | <0.001 |
| LVRWT | 0.3±0.06 | 0.3±0.03 | 0.83 |
| Time after BAV [months] | 101.2±57.7 | 126.7±71.42 | 0.26 |
| LVEF before BAV [%] | 68.9±11.89 | 67.7±9.29 | 0.76 |

Data expressed as mean±SD or percentage.

AR, aortic regurgitation; PG$_{max}$, instantaneous peak systolic Doppler gradient across aortic valve; other abbreviations–see in Table 1.

**Table 3. Strain and strain rate values of left ventricle.**

| Parameter | Patients (n = 40) | Control group (n = 62) | *P* value |
|---|---|---|---|
| **GLS [%]** | -19.7±2.22 | -22.3±1.5 | <0.001 |
| LS—basal segments [%] | -16.4±2.24 | -19.0±1.85 | <0.001 |
| LS—middle segments [%] | -19.7±2.59 | -22.3±1.82 | <0.001 |
| LS—apical segments [%] | -22.9±3.34 | -25.7±2.01 | <0.001 |
| **GCS [%]** | -22.8±1.18 | -22.7±1.37 | 0.66 |
| CS—basal segments [%] | -20.1±1.76 | -20.0±1.94 | 0.90 |
| CS—middle segments [%] | -22.9±1.37 | -22.9±2.51 | 0.94 |
| CS—apical segments [%] | -25.4±2.18 | -25.1±2.06 | 0.44 |
| **GLSr [1/s]** | -0.89±0.15 | -1.04±0.12 | <0.001 |
| LSr—basal segments [1/s] | -0.78±0.19 | -0.98±0.13 | <0.001 |
| LSr—middle segments [1/s] | -0.90±0.14 | -1.03±0.12 | <0.001 |
| LSr—apical segments [1/s] | -1.01±0.21 | -1.10±0.17 | 0.005 |
| **GCSr [1/s]** | -1.39±0.16 | -1.39±0.19 | 0.98 |
| CSr—basal segments [1/s] | -1.34±0.2 | -1.28±0.23 | 0.19 |
| CSr—middle segments [1/s] | -1.29±0.22 | -1.34±0.26 | 0.32 |
| CSr—apical segments [1/s] | -1.53±0.27 | -1.54±0.26 | 0.88 |

Data expressed as mean+SD.

GLS, global longitudinal strain; LS, longitudinal strain; GCS, global circumferential strain; CS, circumferential strain; GLSr, global longitudinal strain rate; LSr, longitudinal strain rate; GCSr, global circumferential strain rate; CSr, circumferential strain rate.

apical segments were significantly lower in the study group compared with the control group (Table 3). Reduced GLS (higher than –19.31%) was found in 19 patients (47.5%).

GLS in the study group showed a significantly positive correlation with the presence of LVEH (r = 0.5, *P*<0.001), PG$_{max}$ (r = 0.46, *P* = 0.002), LVMI (r = 0.46, *P*<0.001), LV RWT (r = 0.2, *P* = 0.003) and LVEDd (r = 0.37, *P*<0.001) (Fig 2).

The correlation between GLS measurements and the presence of grade ≥2 AR was borderline significant (r = 0.29, *P* = 0.05). No correlation was found between GLS and the time since BAV, the BSA value or the study patient age. Also, no correlation was found between GLS and the following parameters of systolic function assessed by two-dimensional echocardiography and TDI: pre-BAV LVEF, LVEF at most recent scan, s' spt and s' lat.

The values of s' spt and s' lat did not differ significantly between the patients with reduced GLS and the patients with normal GLS (*P* = 0.81 and *P* = 0.82, respectively) (Table 4).

In the subgroup of patients with LVEH, GLS and strain in basal, middle and apical segments were significantly lower compared with the non-LVEH subgroup (Table 5).

We found no differences between the type of predominant valvular defect (stenosis/regurgitation/mixed nature) and the change in GLS (*P* = 0.84).

**Analysis of circumferential strain.** GCS and strain in the basal, middle and apical segments did not differ significantly compared with the control group (Table 3). None of the patients in the study group had a GCS reduction of higher than –19.93%.

No significant correlations were observed between GCS and the presence of LVEH, the presence of grade ≥ 2 AR, PG$_{max}$, LVMI, LV RWT, LVEF at most recent scan, pre-BAV LVEF, patient age, or time since BAV.

In the subgroup of patients with LVEH, GCS and strain in the basal and middle segments were lower compared with the non-LVEH subgroup. In the apical segments, on the other

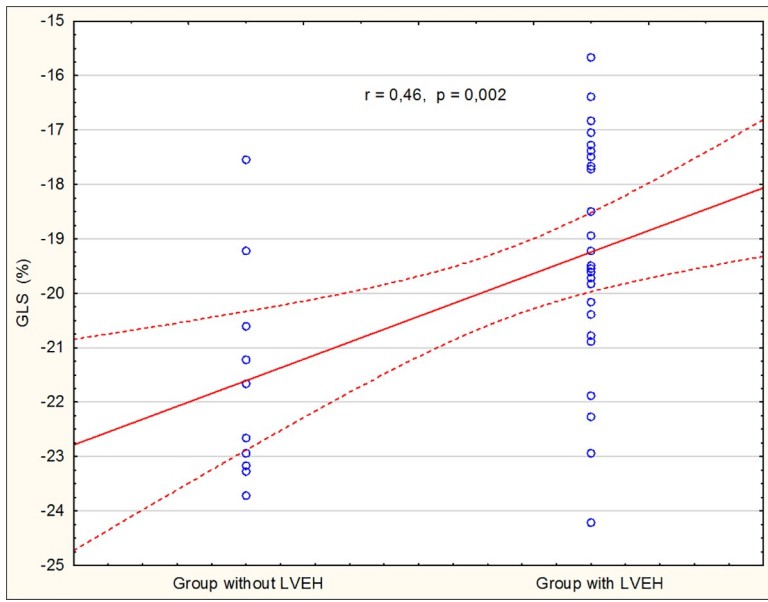

**Fig 2. The correlation of GLS with the existence of LVEH.** GLS, global longitudinal strain; LVEH, left ventricular eccentric hypertrophy.

**Table 4. Risk factors for left ventricular systolic dysfunction defined as GLS $> -19.31\%$ and GLSr $> -0.8$/s in the study group.**

| Parameter | Patients with reduced GLS (n = 19) | Patients with normal GLS (n = 21) | P value |
|---|---|---|---|
| with LVEH/without LVEH | 89.5%/10.5% | 61.9%/38.1% | 0.04 |
| PG$_{max}$ [mmHg] | 37.8±13.31 | 29.5±8.63 | 0.02 |
| AR grade $< 2$/AR grade $\geq 2$ | 21.1%/78.9% | 33.3%/66.7% | 0.38 |
| LVMI [g/m$^{2.7}$] | 49.7±15.32 | 50.1±21.24 | 0.94 |
| LVRWT | 0.31±0.06 | 0.29±0.04 | 0.14 |
| LVEDd Z–score | 2.2±1.87 | 2.3±2.27 | 0.85 |
| LVEF [%] | 64.1±2.82 | 64.0±2.26 | 0.90 |
| S' spt [cm/s] | 7.0±1.29 | 7.0±1.17 | 0.95 |
| S' lat [cm/s] | 8.2±2.04 | 8.0±2.24 | 0.84 |
| LVEF before BAV [%] | 71.7±10.98 | 65.9±10.9 | 0.10 |
| Time after BAV [months] | 113.3±56.25 | 102.4±66.86 | 0.58 |
| **Parameter** | **Patients with reduced GLSr (n = 14)** | **Patients with normal GLSr (n = 26)** | **P value** |
| with LVEH/without LVEH | 100%/0% | 61.5%/38.5% | 0.007 |
| PG$_{max}$ [mmHg] | 43.9±7.11 | 27.8±11.53 | $<0.001$ |
| AR grade $< 2$/AR grade $\geq 2$ | 78.6%/21.4% | 69.2%/30.8% | 0.52 |
| LVMI [g/m$^{2.7}$] | 52.5±19.52 | 48.6±16.61 | 0.53 |
| LVRWT | 0.32±0.04 | 0.29±0.06 | 0.05 |
| LVEDd Z-score | 2.3±2.17 | 2.2±1.91 | 0.89 |
| LVEF [%] | 64.5±2.4 | 63.8±3.2 | 0.44 |
| S' spt [cm/s] | 6.7±1.3 | 7.2±1.01 | 0.29 |
| S' lat [cm/s] | 8.0±2.15 | 8.1±2.15 | 0.89 |
| LVEF before BAV [%] | 69.2±11.74 | 68.3±10.52 | 0.81 |
| Time after BAV [months] | 122.9±63.34 | 99.3±56.95 | 0.25 |

Data expressed as mean±SD or percentage. Abbreviations–see in Table 1.

**Table 5. Strain and strain rate in subgroups: With and without left ventricular eccentric hypertrophy.**

| Parametr | Patients with LVEH (n = 30) | Patients without LVEH (n = 10) | *P* value |
|---|---|---|---|
| **GLS** [%] | -19.1±1.93 | -21.6±1.99 | <0.001 |
| LS—basal segments [%] | -15.7±2.08 | -18.1±1.63 | 0.001 |
| LS—middle segments [%] | -19.3±2.15 | -21.4±3.3 | 0.04 |
| LS—apical segments [%] | -22.1±3.29 | -25.4±2.07 | 0.004 |
| **GCS** [%] | -22.6±1.23 | -23.3±0.84 | 0.09 |
| CS—basal segments [%] | -19.9±1.55 | -20.7±2.23 | 0.17 |
| CS—middle segments [%] | -22.8±1.27 | -23.3±1.63 | 0.30 |
| CS—apical segments [%] | -25.2±2.4 | -26.0±1.09 | 0.33 |
| **GLSr** [1/s] | -0.85±0.15 | -1.01±0.11 | 0.004 |
| LSr—basal segments [1/s] | -0.73±0.19 | -0.91±0.17 | 0.014 |
| LSr—middle segments [1/s] | -0.87±0.13 | -0.99±0.13 | 0.021 |
| LSr—apical segments [1/s] | -0.96±0.21 | -1.13±0.13 | 0.02 |
| **GCSr** [1/s] | -1.39±0.16 | -1.39±0.16 | 0.91 |
| CSr—basal segments [1/s] | -1.32±0.2 | -1.40±0.2 | 0.29 |
| CSr—middle segments [1/s] | -1.28±0.21 | -1.32±0.25 | 0.70 |
| CSr—apical segments [1/s] | -1.56±0.2 | -1.43±0.35 | 0.20 |

Data expressed as mean±SD. Abbreviations–see in Table 3.

hand, circumferential strain was higher compared with the non-LVEH subgroup. However, these differences were not statistically significant (Table 5).

The percentage of patients with LVEH was significantly higher in the subgroup with reduced GLS compared to the subgroup with normal GLS. Also $PG_{max}$ was significantly higher in the subgroup patients with reduced GLS compared to the subgroup with normal GLS (Table 4). No significant differences were demonstrated for the other parameters listed in Table 4.

To identify predictors of reduced GLS the variables: present of eccentric hypertrophy, $PG_{max}$, AR grade, LVMI, LVRWT, LVEDd, LVEF, s' spt, s' lat, LVEF before BAV, time after BAV were all entered into logistic regression analysis. This analysis confirmed only the association with LVEH (OR 6.9; 95% CI: 1.37–12.5; *P* = 0.045).

**Analysis of strain rate.** GLSr and strain rate in the basal, middle and apical segments were significantly lower in the group of post-BAV patients compared with the children in the control group (Table 3). A reduced GLSr (higher than –0.8/s) was found in 14 patients (35%).

GLSr in the study group showed a significantly positive correlation with the presence of LVEH (r = 0.34, *P* = 0.02), $PG_{max}$ (r = 0.53, *P*<0.001) and LV RWT (r = 0.31, *P* = 0.04). No significant correlations were found between GLSr and: LVMI, pre-BAV LVEF, LVEF at most recent scan, BSA, presence of grade ≥2 AR, time since BAV, patient age, s' spt and s' lat.

In the subgroup of patients with LVEH, GLSr and strain rate in the basal, middle and apical segments were significantly lower compared with the non-LVEH subgroup (Table 5). All the patients with reduced GLSr were found to have LVEH. The values of $PG_{max}$ and LV RWT were significantly higher in patients with reduced GLSr compared to the subgroup with normal GLSr (*P*<0.001 and *P* = 0.048, respectively) (Table 4).

None of the patients in the study group had a GCSr reduction. GCSr and strain rate in the basal, middle and apical segments did not differ significantly between the study group and the control group (Table 3) and between the subgroup of patients with LVEH and the non-LVEH subgroup. No significant correlations were observed between GCSr and the factors listed in the Table 4.

## Intra- and interobserver variability

Intraobserver variability of GLS, GCS, GLSr and GCRs was 4.3; 5.6; 6.0; 5.6%, respectively.

The corresponding interobserver values were 12.4; 10.2; 16.0; 12.6%, respectively. The intra-class correlation coefficient (ICC) for intraobserver variability regarding GLS, GCS, GLSr and GCRs were 0.98(95% CI 0.98–0.99); 0.95(95% CI 0.93–0.96); 0.99(95% CI 0.98–0.99); 0.96 (95% CI 0.95–0.97), respectively. The corresponding ICC values for interobserver variability were 0.85(95% CI 0.81–0.89); 0.81(95% CI 0.74–0.85); 0.9(95% CI 0.88–0.93); 0.82(95% CI 0.76–0.86), respectively.

## Discussion

Information on myocardial dysfunction in children undergoing BAV for AS is limited. To the best of our knowledge, this is the first study to assess left ventricle using 2D STE in children during long-term follow-up after BAV. There are only two studies evaluating the left ventricle by STE in children after BAV for AS. The first one concerns intermediate- term results in 37 children and the second one concerns results in 27 infants shortly after BAV procedure. Our study has shown LV longitudinal systolic function impairment and its dependence on LV hypertrophy as a result of both increased pre- and afterload in this group of patients. We have also demonstrated reduced global and regional longitudinal strain and longitudinal strain rate with normal values of circumferential strain and circumferential strain rate.

2D STE allows to precisely assess myocardial function. It is an accurate and reproducible method that is independent of the angle of incidence of the ultrasonic beam, allows to differentiate between active and passive strain, and allows a complete assessment of regional and global function in three directions. In contrast, TDI is a method that is dependent on the angle of incidence, susceptible to noise and less accurate, and it allows to assess only a limited fragment of tissue [11,12].

When subjected to an increased preload caused by AR [13], and/or an increased afterload caused by residual systolic pressure gradient across the aortic valve [14–16] the left ventricle undergoes remodelling (eccentric hypertrophy), which was observed in most of our patients. In the initial phase, the increase of wall thickness in particular is a beneficial adaptive mechanism which, according to Laplace's law, contributes to the reduced stress within the ventricular wall. Over time, however, it turns into an unfavourable phenomenon that leads to impaired ventricular function [17,18].

Leonardi et al. [19] and Singh et al. [20] demonstrated an increased left ventricular mass in addition to the increased left ventricular end-diastolic dimension in children with mixed aortic valve disease. Furthermore, in a group of children and young adults, Hill et al. [21] demonstrated a higher left ventricular end-diastolic dimension in patients with mixed aortic valve disease compared to the group with isolated AS.

Based on our preliminary findings and on the theoretical and practical premises, we analysed the effects of the existing LV remodelling on LV systolic function as assessed by 2D STE, conventional methods, and tissue Doppler imaging.

Left ventricular systolic function assessed by conventional method (ejection fraction) was normal in all our patients, including those who presented with a reduced ejection fraction prior to BAV. Partial consistency between PW-TDI and 2D STE has been demonstrated for the assessment of longitudinal systolic function of the myocardium. While a significant reduction was found for lateral mitral annulus velocity (s' lat), the velocities of the medial portion of the annulus (s' spt) were normal compared to the control group.

Notably, ejection fraction is a parameter that is significantly dependent on LV loading condition and therefore its ability to detect systolic dysfunction may be limited. PW-TDI, on the other hand, mainly reflects longitudinal systolic function of the myocardium.

Also, the analyses conducted in adult patients suggest that the detection of impaired LV systolic function by conventional echocardiography (EF) indicates the irreversibility of the completed changes and therefore contributes to worse treatment outcomes and increased mortality [13].

Left ventricular systolic function impairment expressed as reduced GLS and regional strain was present in nearly half of our patients, while that expressed as GLSr and regional strain rate was identified in 35% of the patients. We showed that the only factor that impaired GLS was LVEH, whose presence increased the risk of longitudinal myocardial dysfunction by nearly seven-fold. These findings point to injury within the subendocardial layer, which is responsible for longitudinal LV strain, while no circumferential myocardial dysfunction was identified.

Only two available studies showed the assessment of myocardial function in children during the intermediate- and short-term follow-up after BAV. In the first one, Marcus et al. [22] observed an improvement in myocardial strain but not its normalisation. They demonstrated that the reduced pre-BAV left ventricular longitudinal and circumferential strains increased significantly at 6 months post-procedure but failed to improve over the 2.5 years of further follow-up, while radial strain did normalise. In the second one, Ankola et al. demonstrated normalisation of reduced LV longitudinal strain in small infants shortly after BAV due to severe AS, while the circumferential strain was not affected neither before nor after the procedure [23]. They also found no abnormality of myocardium function in circumferential direction, which was consistent with our study.

The factors responsible for the unfavorable consequences of LV remodeling remain unclear. The impairment of longitudinal strain and strain rate combined with the simultaneous preservation of normal values of circumferential strain and stain rate in our study suggests that the subendocardial layer of longitudinal muscle fibres is, by virtue of their location, more susceptible to pathological changes than the middle and subepicardial layers of fibres, which run in oblique and transverse directions. Under conditions of increased pre- and afterload, especially with increased LVEDd, the increased left ventricular wall stress impairs the perfusion in the subendocardial region distant from the epicardial coronary vessels. In accordance to Laplace's law, left ventricular wall stress increases with increased left ventricular end-diastolic dimension. This is confirmed by the presence of LVEH in 75% of our patients.

GLS also showed significant correlation with LVEDd and LVEH.

Some studies in adults patients also show a relation between longitudinal strain impairment and fibrosis measured by gadolinium enhanced cardiac magnetic resonance imaging [24,25].

There are several published studies in literature that were conducted in patients with AR which did not develop as a complication of interventional treatment. Lowenthal et al. [26] found GLS to be impaired in children with moderate to severe AR. They also showed that the reduction of GLS below −15.3% and that of GLSr below −0.79/s indicated the persistence of impaired left ventricular function after the aortic valve surgery. In addition, circumferential strain was no different in the quoted study, similar to this current study. Leonardi et al. [19] demonstrated reduced GLS and elevated GCS in older children and young adults with aortic stenosis combined with regurgitation. Similar findings have been reported by Ewe et al. [27], Mizariene et al. [28] and Iida et al. [29] in adult patients with isolated AR. Reduced longitudinal strain and lower compliance of the middle and subendocardial layers of the left ventricular myocardium in response to the pathological effects of increased pre- and afterload have also been reported by other authors [26,30].

Our observations suggest that the initial functional changes occur in the endocardium, where the longitudinal course of the muscle fibres predominates. The preserved normal values of circumferential strain and strain rate may play a compensatory role in maintaining LVEF in the presence of impaired global longitudinal stain. Therefore, the assessment of longitudinal LV systolic function by measuring global longitudinal strain by 2D STE is an appropriate indicator for detecting subclinical systolic dysfunction in patients after BAV. 2D STE is more sensitive than the evaluation of tissue velocities or that the conventional one- and two-dimensional echocardiographic methods. Importantly, longitudinal systolic function impairment identified in patients more than 3 years after the procedure may suggest that pathological changes (e.g. fibrosis) are becoming permanent at least in the subendocardial layers of the myocardium.

These findings may additionally inform the timing of aortic valve surgical treatment for AR and/or the presence of a residual pressure gradient after previous BAV.

In conclusion, 2D STE allows to detect LV systolic function impairments in patients with normal parameters on conventional echocardiography.

## Conclusions

Patients who have undergone balloon aortic valvuloplasty present subclinical left ventricular systolic impairment in the long-term follow-up which is associated with the presence of left ventricular remodelling. Longitudinal strain is the most sensitive marker of left ventricular systolic impairment in this group of patients.

## Supporting information

**S1 Data.**
(XLS)

## Author Contributions

**Conceptualization:** Krzysztof Godlewski, Bożena Werner.

**Data curation:** Krzysztof Godlewski, Paweł Dryżek, Elżbieta Sadurska.

**Formal analysis:** Krzysztof Godlewski.

**Investigation:** Krzysztof Godlewski, Paweł Dryżek, Elżbieta Sadurska, Bożena Werner.

**Methodology:** Krzysztof Godlewski.

**Supervision:** Bożena Werner.

**Validation:** Krzysztof Godlewski, Bożena Werner.

**Writing – original draft:** Krzysztof Godlewski.

**Writing – review & editing:** Krzysztof Godlewski, Bożena Werner.

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
