## [Decision Letter · Decision Letter 0]

15 Dec 2020

PONE-D-20-36499

Left Ventricular Systolic Function Impairment in Children after Balloon Valvuloplasty for Congenital Aortic Stenosis Assessed by 2D Speckle Tracking Echocardiography

PLOS ONE

Dear Dr. Werner,

Thank you for submitting your manuscript to PLOS ONE. After careful consideration, we feel that it has merit but does not fully meet PLOS ONE’s publication criteria as it currently stands. Therefore, we invite you to submit a revised version of the manuscript that addresses the points raised during the review process.

We look forward to receiving your revised manuscript.

Kind regards,

YIRU GUO, M.D., F.A.H.A.

Academic Editor

PLOS ONE

Journal Requirements:

2.Thank you for including your ethics statement:

"Bioethical Committee of the Medical University of Warsaw KB/45/2014, written consent.".

i) Please amend your current ethics statement to confirm that your named institutional review board or ethics committee specifically approved this study.

ii) Once you have amended this/these statement(s) in the Methods section of the manuscript, please add the same text to the “Ethics Statement” field of the submission form (via “Edit Submission”).

3. You indicated that you had ethical approval for your study. In your Methods section, please ensure you have also stated whether you obtained consent from parents or guardians of the minors included in the study or whether the research ethics committee or IRB specifically waived the need for their consent.

4.We note that you have indicated that data from this study are available upon request. PLOS only allows data to be available upon request if there are legal or ethical restrictions on sharing data publicly. For information on unacceptable data access restrictions, please see http://journals.plos.org/plosone/s/data-availability#loc-unacceptable-data-access-restrictions.

Reviewers' comments:

Reviewer's Responses to Questions

**Comments to the Author**

1. Is the manuscript technically sound, and do the data support the conclusions?

Reviewer #1: Yes

Reviewer #2: Yes

2. Has the statistical analysis been performed appropriately and rigorously? 

Reviewer #1: Yes

Reviewer #2: Yes

3. Have the authors made all data underlying the findings in their manuscript fully available?

Reviewer #1: Yes

Reviewer #2: Yes

4. Is the manuscript presented in an intelligible fashion and written in standard English?

Reviewer #1: Yes

Reviewer #2: Yes

5. Review Comments to the Author

Reviewer #1: Left Ventricular Systolic Function Impairment in Children after Balloon Valvuloplasty for Congenital Aortic Stenosis Assessed by 2D Speckle Tracking Echocardiography

This study attempts to investigate left ventricular remodeling after balloon valvuloplasty for aortic stenosis and discovered lower longitudinal strain and strain rate for these patients after a follow-up period compared to control population. The authors appropriately address an important question that aids clinicians’ understanding of the ventricle in patients with addressed aortic stenosis and may have data that have long term implications for these patients. I have some important concerns about the study design, interpretation of data results, and the authors’ conclusions. The patients studied have varying degrees of aortic stenosis, aortic insufficiency or both. The study population is not homogenous with varying amounts of preload from aortic insufficiency and afterload from aortic stenosis; how can the reader tell what variable affects strain more with this mixture of patients. The authors should ideally split the population as a subanalysis to see if the same statistical results hold true by looking at mostly aortic insufficiency, mostly aortic stenosis, and mixed disease. The follow-up time period needs to be clearly stated and is not presented in the abstract – what does long-term follow-up mean? The reader needs a clear definition with years (or months) and a range or standard deviation. Another important point is that the strain values are not that different from established normal; so are they truly depressed? I would argue that they are lower by paired comparison with the control group but not necessarily lower than established normal, unless the authors show that with prior established normals for the strain package the study includes. Why did the authors have 40 study and a different number of control patients? Were the comparisons to controls not paired? Below are additional comments I have in order of appearance within the authors’ work:

1. Abstract – long term – this needs to be clearly defined in months with a range/sd; see comment above

2. Abstract – the GLS should be just one decimal point (-19.69 should be -19.7); I don’t think strain is accurate to 4 significant figures

3. Abstract – what other variables besides GLS were in the multivariable model; this should be included in the abstract too

4. Abstract – the authors describe “left ventricular systolic dysfunction” in the study patients – I would argue that the strain may be within the normal limits. I would state lower strain values than the control patients in the study; please see my prior comment above

5. Abstract – balloon is mispelled

6. Introduction – The authors state: “The conventional two-dimensional echocardiography (left ventricular ejection fraction [LVEF] assessment) fails to reveal LV systolic function impairment for a long duration of the disease.” Do we know for sure that there is systolic function impairment? I would argue that the conventional measurements show presumed normal function or even normal FS or EF.

7. Introduction – The authors state that the diminished LVEF in adults “already indicates an advanced degree of LV dysfunction.” How do we know this? What is a marker of this?

8. Methods – how do we know that the control group is healthy? Were these volunteers? Were these patients that happened to come for indications that were benign and discovered to have innocuous disease? Why are there not an equal number of control patients? Did the controls not get matched to the age/size of the study patients?

9. Table 1 – The authors have a 0,7 when it should be a decimal place. 0.7 should have 2 significant figures in the p-value section like the other p-values. Other than IVSd, LVPWd, LVRWT, the other values should have only one decimal place.

10. Methods – LVRWT<0,44 should be 0.44 and did the authors use a mass/volume ratio to detect eccentric hypertrophy? Is there a reference for LVRWT the authors can supply to justify the eccentric hypertrophy definition?

11. Methods – The authors state “By using continuous-wave Doppler (CW-Doppler) and color Doppler an assessment of the LV-Ao instantaneous peak systolic Doppler gradient (PGmax) and the grade AR (on a 4-point grading scale) were made.” I believe an assessment “was” made not “were” made.

12. Methods – pulse-wave should be pulsed-wave Doppler (line 95)

13. Methods – line 107, did the authors or the software reject poorly tracking segments?

14. Methods – line 115-116, the authors used the control group to establish cut-off values with a standard deviation. The authors should look at established normal values for strain for the software utilized in this study. If none exist, then the authors should state this.

15. Methods – Statistical Analysis – ICC and interobserver variability is not mentioned here when these comparisons are in the results. How was percentage difference calculated? This makes a significant difference. Did the authors compare percentage difference (100%*(observer1-observer2)/(average of observers) or just the absolute difference such as -17.1% vs -16.1% (1% difference). The first yields a 6% difference while the second is just a 1% difference. I highly recommend that it should be a percentage difference (the first one with 6%) since that clearly tells the variability.

16. Results – The follow-up data after BAV should be given in line 133 after the description of BAV. This way the reader knows precisely how long there has been remodeling.

17. Results – The AR and AS patients overlap and a VENN diagram or something equivalent to show – pure AS, pure AI and mixed disease would be much more helpful

18. Table 2 – the values should have only one decimal point for the LVEH and no LVEH columns. P-value columns are fine except for the LVEF P-value should be two decimals not one.

19. Results – line 163 what does value greater mean. -19% is technically less than -18% but we know for strain that is not the case. I would state the “The magnitude value” or “The absolute value”. The authors should use just 2 decimal places for strain rate and one for strain on line 163 (ie., -19.31 should be -19.3 while -0.8/s should be -0.80/s).

20. Results – line 166 I would replace the word “impaired” with “less”.

21. Results – I strongly urge a subanalysis of GLS for mostly volume loaded left ventricles, mostly pressure loaded left ventricles, and mixed disease to determine if there is a category that is more strongly lower than the normal cut-off values.

22. Table 3 – p-value 0.9 should be two decimal places. The other values for strain should have one decimal place while strain rate two decimal places.

23. Results – line 177: GLS in the study group correlated positively with LVEH. This makes no sense since higher LVEH made the GLS lower. Are the authors suggesting that the strain improves with more aortic stenosis or insufficiency?

24. Table 4/5 – please see prior comments about significant figures/decimal places. The authors should use the abbreviation for LVEH or take out the abbreviation altogether.

25. Results – lines 227-245 – these paragraphs reiterate what is in the table and is redundant; this needs to be summarized into just the most pertinent findings or the reader will get distracted

26. Results – were the interobserver variability results blinded. Were the %differences absolute percent difference or just the difference in strain? Please see my prior comment on reproducibility. The authors report reproducibility on strain. What about strain rate?

27. Discussion – what disease aortic stenosis/regurgitation/mixed disease causes the biggest difference in strain values from the control group.

28. Discussion – line 340 - Notably circumferential strain was no different in the quoted study, similar to this current study.

Reviewer #2: Overall:

This manuscript presents a study that aims to evaluate LV remodeling and systolic function using 2D STE in 59 children during long-term follow-up after BAV for AS. The results presented in this manuscript provides valuable information for the clinical assessment of this important complex pediatric condition.

Title:

“Left Ventricular Systolic Function Impairment in Children after Balloon Valvuloplasty for Congenital Aortic Stenosis Assessed by 2D Speckle Tracking Echocardiography”

The tittle seems appropriate for the manuscript.

Abstract:

The length and content of the abstract are adequate for the manuscript. LVEF should be spelled out in the abstract (other abbreviations are included in the abstract)

Manuscript

Introduction

The length and content of the introduction section is appropriate.

Materials and Methods

The length and content of the methods section is appropriate, however one minor point is, if the authors could not test inter and intra -observer variability, could they provide a reference where it has been tested in the past? It would make the manuscript stronger.

Results

The length and content of the methods section is appropriate.

Discussion

The content, organization and length are appropriate

6. PLOS authors have the option to publish the peer review history of their article (what does this mean?). If published, this will include your full peer review and any attached files.

Reviewer #1: No

Reviewer #2: No

---

## [Author Response · Author response to Decision Letter 0]

29 Jan 2021

Reviewer 1: I have incorporated all of your suggestions into my revision. They were very helpful. Thank You.

Reviewer 2: I have incorporated all of your suggestions into my revision. Thank you for your help.

---

## [Decision Letter · Decision Letter 1]

8 Mar 2021

Left Ventricular Systolic Function Impairment in Children after Balloon Valvuloplasty for Congenital Aortic Stenosis Assessed by 2D Speckle Tracking Echocardiography

PONE-D-20-36499R1

Dear Dr. Werner,

We’re pleased to inform you that your manuscript has been judged scientifically suitable for publication and will be formally accepted for publication once it meets all outstanding technical requirements.

Kind regards,

YIRU GUO, M.D., F.A.H.A.

Academic Editor

PLOS ONE

Additional Editor Comments (optional):

Reviewers' comments:

Reviewer's Responses to Questions

**Comments to the Author**

1. If the authors have adequately addressed your comments raised in a previous round of review and you feel that this manuscript is now acceptable for publication, you may indicate that here to bypass the “Comments to the Author” section, enter your conflict of interest statement in the “Confidential to Editor” section, and submit your "Accept" recommendation.

Reviewer #1: All comments have been addressed

Reviewer #2: All comments have been addressed

2. Is the manuscript technically sound, and do the data support the conclusions?

Reviewer #1: Yes

Reviewer #2: Yes

3. Has the statistical analysis been performed appropriately and rigorously? 

Reviewer #1: Yes

Reviewer #2: Yes

4. Have the authors made all data underlying the findings in their manuscript fully available?

Reviewer #1: Yes

Reviewer #2: Yes

5. Is the manuscript presented in an intelligible fashion and written in standard English?

Reviewer #1: Yes

Reviewer #2: Yes

6. Review Comments to the Author

Reviewer #1: The authors have adequately addressed my concerns on this paper and here a few more suggestions:

1. In the discussion, the authors should probably acknowledge that the comparisons among aortic stenosis, aortic regurgitation, and mixed diseases yielded no difference; however, the sample sizes were relatively small in each subcategory which may have affected this comparison.

2. Although the graph for gls and lveh mathematically shows a "positive" correlation given negative values, I recommend reiterating that the more negative GLS values correspond with higher strain when explaining these findings in the Results section too. Positive correlation implies an "improvement" in function when, in fact, it's the opposite.

Reviewer #2: The authors have satisfactorily addressed the comments from the reviewers and in my opinion this manuscript should be accepted for publication.

7. PLOS authors have the option to publish the peer review history of their article (what does this mean?). If published, this will include your full peer review and any attached files.

Reviewer #1: No

Reviewer #2: No

---

## [Editor Report · Acceptance letter]

21 Apr 2021

PONE-D-20-36499R1 

Left Ventricular Systolic Function Impairment in Children after Balloon Valvuloplasty for Congenital Aortic Stenosis Assessed by 2D Speckle Tracking Echocardiography 

Dear Dr. Werner:

I'm pleased to inform you that your manuscript has been deemed suitable for publication in PLOS ONE. Congratulations! Your manuscript is now with our production department. 

Kind regards, 

on behalf of

Dr. YIRU GUO 

Academic Editor

PLOS ONE